# Optimization Method of the Solvothermal Parameters Using Box–Behnken Experimental Design—The Case Study of ZnO Structural and Catalytic Tailoring

**DOI:** 10.3390/nano11051334

**Published:** 2021-05-19

**Authors:** Zoltán Kovács, Csanád Molnár, Urška Lavrenčič Štangar, Vasile-Mircea Cristea, Zsolt Pap, Klara Hernadi, Lucian Baia

**Affiliations:** 1Research Group of Environmental Chemistry, Institute of Chemistry, University of Szeged, Tisza Lajos krt. 103, H-6720 Szeged, Hungary; kovacs.zoltan.ubb@gmail.com (Z.K.); Molnar.Csanad@stud.u-szeged.hu (C.M.); 2Faculty of Physics, Babeș-Bolyai University, Str. Mihail Kogălniceanu 1, RO-400084 Cluj-Napoca, Romania; lucian.baia@phys.ubbcluj.ro; 3Department of Applied and Environmental Chemistry, University of Szeged, Rerrich tér 1, H-6720 Szeged, Hungary; 4Faculty of Chemistry and Chemical Technology, University of Ljubljana, Večna pot 113, 1000 Ljubljana, Slovenia; Urska.Lavrencic.Stangar@fkkt.uni-lj.si; 5Faculty of Chemistry and Chemical Engineering, Babeș-Bolyai University, Str. Arany János 11, RO-400028 Cluj-Napoca, Romania; mircea.cristea@ubbcluj.ro; 6Centre of Nanostructured Materials and Bio-Nano Interfaces, Institute for Interdisciplinary Research on Bio-Nano-Sciences, Treboniu Laurian 42, RO-400271 Cluj-Napoca, Romania; 7Institute of Physical Metallurgy, Metal Forming and Nanotechnology, University of Miskolc, HU-3515 Miskolc-Egyetemváros, Hungary

**Keywords:** Box–Behnken design (BBD), ZnO, photocatalysis, solvothermal crystallization, optimization

## Abstract

ZnO photocatalysts were synthesized via solvothermal method and a reduced experimental design (Box–Behnken) was applied to investigate the influence of four parameters (temperature, duration, composition of the reaction mixture) upon the photocatalytic activity and the crystal structure of ZnO. The four parameters were correlated with photocatalytic degradation of methyl orange and the ratio of two crystallographic facets ((002) and (100)) using a quadratic model. The quadratic model shows good fit for both responses. The optimization experimental results validated the models. The ratio of the crystal facets shows similar variation as the photocatalytic activity of the samples. The water content of the solvent is the primary factor, which predominantly influence both responses. An explanation was proposed for the effect of the parameters and how the ratio of (002) and (100) crystal facets is influenced and its relation to the photocatalytic activity. The present research laconically describes a case study for an original experimental work, in order to serve as guideline to deal with such complicated subjects as quantifying influence of synthesis parameters upon the catalytic activity of the obtained ZnO.

## 1. Introduction

In the last few decades, various semiconductor oxides have been employed as photocatalysts (such as TiO_2_, ZnO, Fe_2_O_3_, WO_3_, BiVO_4_) [1]. Amongst these oxide materials, zinc-oxide (ZnO) has received tremendous attention, because of its versatility in many different fields of application, such as field emission displays [2], varistors [3], sensors [4], piezoelectrical component, and most extensively, has been studied in the field of heterogenous catalysis for wastewater treatment [5,6,7]. The basis of the photocatalytic phenomenon lies within the semiconductor nature of the oxide catalyst. When a semiconductor is irradiated by electromagnetic radiation with an energy quantum equal or higher than its band-gap energy, an electron from the valence band is transferred to the conduction band, leaving behind a positive charged hole. Subsequently these separated charges can interact separately with other molecules inducing redox reactions or generate highly reactive radicals with dissolved oxygen (•OH, •O^2−^), which contribute to the mineralization process of organic pollutants present in wastewater [8]. 

Nowadays the focus concerning ZnO is turned towards material science related aspects, where it is well-known that the structural, morphological, optical, and surface proprieties are the primary qualities, which determine the catalytic activity [9]. Thus, the research trends of ZnO photocatalyst are currently focused on identifying the mentioned properties and further optimizing them by different synthesis methods (sol-gel [10], low-temperature chemical precipitation [11], electrochemical deposition [12], chemical vapor deposition [13], etc.). The main challenge is to identify the parameters that could positively influence these properties and implicitly help to the design of the (photo) catalyst, in general or specifically, for the photodegradation of organic compounds in order to get a more efficient remediation of the contaminated water [14]. Solvothermal techniques for the synthesis of pure ZnO powders is most reported with the preparation of a liquid reaction mixture of a designated precursor. The crystallization process is promoted in a single step thermal treatment in a sealed laboratory autoclave, followed by a washing and drying step. The role of the solvothermal treatment is to influence the crystallizations’ kinetics and energetic aspects to promote an existing reaction towards the formation of ZnO crystals [15].

For the preparation of ZnO, several types of solvents, precursors, and additives can be used, which overall determines the possible reaction routes. The optimal synthetic conditions used to maximize the photocatalytic activity for the solvothermal preparation of ZnO nanoparticles has already been attempted several times [16,17]. A. Famengo et al. have reported that ZnO prepared in organic solvents, but especially in ethanol, leads to highly active photocatalysts than in water [18]. Also, several precursors have been studied to prepare ZnO using solvothermal method, such as inorganic salts (zinc-acetate [19,20], -nitrate [6,21]), as well as organic complexes (zinc-acetylacetonate [15], -oximate [22], -alkoxides [23]). One of the most investigated crystallization mechanisms is the controlled hydrolyzation of zinc-acetylacetonate that predominantly consists of the hydroxylation of the central Zn^2+^ ion of the compound in mixtures of water and different organic solvents [24]. By this simple approach there were several successful attempts in the literature, which proved that the crystallization of the semiconductor from similar reaction mixture is directly responsible for the photocatalytic activity of the obtained ZnO samples, by manipulating the structure of the crystals and their morphology. In a typical solvothermal synthesis route involving the usual laboratory scale autoclaves, without any internal sensors, there is no possibility to monitor the crystallization process in the sealed autoclave. Therefore, the information is gathered only from the preliminary reaction mixture and the measurements carried out upon the obtained materials after the end of the thermal treatment step [18]. 

Mathematically speaking, this situation resembles very much with that of a black box system than a white box system, because between the input and output variables the internal relationships we could only assume the phenomenologically corresponding mathematics. Therefore, we could only observe the overall sum of an empirical trend [24,25,26]. Consequently, the search for the simplest and most efficient control upon the photocatalytic activity of ZnO obtained through solvothermal synthesis has become a scientific curiosity and lately an annoyance. Most of the literature attends the description of varying one or two parameters. In most of the cases these singular or binary variable systems can be interpreted with fair accuracy in a linear fashion. Due the many possible interactions between multiple parameters of the synthesis of ZnO, the obstacle to quantify an actual empirical correlation between all primary controllable parameters and photocatalytic activity roots to the fact of high experimental number. By adding new variables to the correlation, the amount of data required to describe the relation for further use to predict new experimental results grows exponentially. This is a requirement to ensure a closely accurate description of their individual and interacting influence with the possibility of deviating from linear fashion behavior. Eventually this renders it a more risky and more consuming approach, in terms of resources and time. An exemplary case for such situation is the work of Jacek Wojnarowicz et al. Their results showed that by adjusting the water content of the solvent (ethylene-glycol) in range of 1–4% in microwave solvothermal synthesis of ZnO, the particle size can be easily controlled between 20 and 120 nm [19,20]. These types of research offer great results and should further increase their parameter range, but that would require a higher number of experiments, which is laborious and time consuming. A similar work also proves a quantifiable relationship between activity, size, and solvothermal synthesis parameters, but with a very extensive spectrum of experimental analysis [27]. 

In order to combat the above-mentioned disadvantages one approach would be to minimize the experimental and measurement requirements by fractional factorial experimental designs, which exploit the sparsity-of-effects principle to reduce a full factorial design. Fractional factorial design applied to examine synthesis parameter effects upon catalytic properties has proven to be a straightforward tool to find the optimum conditions to achieve a higher activity of the respective catalyst without using an extensive investigation (X-ray Photoelectron Spectroscopy (XPS), BET specific surface measurements etc.) of specific material properties related to the activity. Poortavasoly et al. have already successfully optimized the synthesis parameters of ZnO to obtain the maximum activity of composite structures using Taguchi design [28]. One of the most used experimental design is Box–Behnken design (BBD), which yields fairly stable results in case of systems with 4 up to 7 input parameters. As the precedence of already existing data confirms BBDs have been proven successful to determine empirical kinetic models for photocatalysis of several organic dyes, pharmaceuticals, and typical model pollutants (phenol, gallic acid, etc.), photochemical reactions or to optimize structural proprieties of semiconductors, it was not used in the case of pure ZnO synthesized via solvothermal method to optimize photocatalytic efficiency, even though such a design would require less than 40% of experiments of the initial full factorial design and less time and resource consuming material investigations techniques. Hence, in the present work, the 4 most generally controllable parameters of the solvothermal synthesis of ZnO have been investigated in a 3 level BBD and analysis of variance (ANOVA) was used to fit a full quadratic model and to subject the parameters to significance tests. In order to prove that experimental designs straightforward can utilized to improve the activity of the catalyst and also to prevent unnecessary time- and resource-consuming investigations (such as XPS, BET).

## 2. Materials and Methods

### 2.1. Reagents and Materials

All chemicals used here were of reagent grade and used without further purification. The materials used for the solvothermal synthesis for ZnO photocatalysts were the following: zinc-acetylacetonate monohydrate (ZnAA_2_, Zn(C_5_H_7_O_2_)_2_·H_2_O, >99%, purchased from Alfa Aesar, Karlsruhe, Germany) as precursor; absolute ethanol (EtOH, 100%, Molar Chemicals, Halásztelek, Hungary) and double distilled ultrafiltered water (MQ, specific conductivity 0.055–0.056 μS/cm) as solvents. The organic compound used as model pollutant for photodegradation was methyl orange (MO) (99.99%, VWR International, Debrecen, Hungary). A reference catalyst was used Evonik Aeroxide P25 (Essen, Germany).

### 2.2. Solvothermal Synthesis of ZnO 

In each experiment we prepared a reaction mixture by dissolving a certain amount of ZnAA_2_ corresponding to a concentration of X_1_ (0.068, 0.136, and 0.204 molar) in a solvent prepared by mixing MQ water and absolute ethanol, corresponding to a certain concentration of ethanol, X_2_ (30, 60, and 90% *v*/*v*). The precursor was first added to pure ethanol and stirred continuously for 40 min using magnetic stirrers at a speed of 500 rpm to ensure solution homogeneity then the corresponding amount of MQ water was added to obtain the desired solvent composition and was stirred for another 20 min. This was followed by the transfer of the reaction mixture into a Teflon lined autoclave, with a 70% active volume, and subjected to a solvothermal treatment at a temperature of X_3_ (90, 140, and 190 °C) with a heating rate of 5 °C min^−1^ for a duration of X_4_ (4, 8, and 12 h). In all the cases we obtained a white precipitate, which was washed several times with ethanol and dried for 12 h at 80 °C. All syntheses were carried out in triplicate and in 2 autoclaves with internal volumes of 150 and 300 mL, respectively. A proposed mechanism for the formation of ZnO from the precursor and solvent mixture is represented in Figure 1.

The obtained crystalline phases were identified by X-ray diffractometry (XRD, Rigaku MiniFlex II diffractometer, Tokyo, Japan) using Cu-K_α_ radiation (λ = 1.5406 Å) equipped with a graphite monochromator, between 29 and 39° to observe the 3 most important and intensive characteristics of the ZnO wurtzite structure: (1 0 0), (0 0 2), (1 0 1) [29]. Mean primary crystallite sizes were calculated applying the Scherrer equation [16].

In order to evaluate a possible correlation between the structure and photodegradation efficiency (PDE), ANOVA analysis was performed for the following values: ratio between the intensities of the identified X-ray diffraction peaks and PDE fitted with a full quadratic function with the solvothermal parameters.

Jasco-V650 UV–vis spectrometer (Jasco, Tokyo, Japan) with an integration sphere (ILV-724) was used to determine the diffuse reflectance spectra (DRS) of catalysts in the wavelength range of 250–800 nm using BaSO_4_ as the reference standard. In order to determine the band-gap energy of the synthesized samples the DRS were transformed using the Kubelka–Munk equation and using Tauc plot representation, respectively, the possible electron transition was calculated by the first derivative of the DRS [11].

The morphology was analyzed by a Hitachi S-4700 Type II scanning electron microscope (Tokyo, Japan).

### 2.3. The Assessment of the Photocatalytic Activity

The photocatalytic efficiency of the ZnO samples was evaluated by the decolorization of MO in aqueous solutions in batch experiments. In a photochemical reactor system under UV-A irradiation (6 ×6 W fluorescent lamps, λ_max_ ≈ 365 nm), the irradiation time was 1 h. For a typical experiment, 130 mL standard solution of 80 µM (C_0_) MO was prepared.

In all experiments the catalyst was added according to a concentration of 1 g∙L^−1^, then the obtained mixture was stirred in a dark environment for 20 min to ensure the adsorption-desorption equilibria. During the decolorization experiment, the suspension was continuously stirred (400 rpm) and, in addition, was purged by air to maintain the dissolved oxygen concentration.

Quantitative analyses of the MO present in the reaction solution during irradiation was carried out by UV-Vis spectroscopy at absorption maximum of 513 nm (using JASCO-V650 spectrophotometer, using the calibration curve presented in Appendix A). Sampling was carried out as following: to ensure adsorption-desorption equilibrium to set in the solution was kept in the dark under stirring, after which samples were taken every 10 min for 1 h. PDE was calculated by the conversion of the MO during photodegradation test, which was computed by using Equation (1).
Y = (C_0_ − C_1_)/C_0_ × 100 [%],(1)
where Y is the conversion, C_0_ is the concentration of MO after 10 min in the dark, and C_1_ denotes for the concentration of MO 40 min of the degradation.

The stability of the best performing sample was investigated through the reuse of the photocatalyst in 3 cycle degradation experiment. One cycle consisted in a degradation experiment described before, in addition of separation of the catalyst from the solution by centrifugation, followed by drying at 40 °C for 12 h. After each cycle the catalyst is reintroduced in a new experimental cycle.

### 2.4. Experimental Design

The Box–Behnken design (BBD) with response surface methodology (RSM) was applied to investigate the influence of 4 major independent variables (molar concentration of the precursor in the reaction mixture, ethanol content of the solvent, temperature, and duration of the crystallization procedure) [17,25,28]. The interaction effects between synthesis variables and their influence on the response (dependent variable) were quantified. Furthermore, the developed prediction model was used to optimize the synthesis conditions for the higher photocatalytic degradation efficiency of the obtained ZnO. 

The Box–Behnken design is a second-order technique based on three-level factorial design (suited for three factors and more), with selected points from a system arrangement [20]. The Box–Behnken design based on RSM was chosen in this study since this design is more effective than the other RSM designs (full factorial designs and central composite design), as it requires a smaller set of experimental data for the case of four independent variables. A visual representation of reduced experimental designs (BBD and central composite design) can be seen in Figure 2. The number of experimental runs required (N) is calculated by Equation (2).
N = 2k (k − 1) + C,(2)
where the number of factors is k and the central point is C. To improve the stability and adequacy of the model, all factors have been adjusted to three levels: −1 (lower), 0 (medial), and 1 (higher) and the central point of the experimental design (with parameter coordinates: precursor concentration 0.136 M, 60% *v*/*v* ethanol-water solvent, 140 °C, 8 h solvothermal treatment) was carried out three times. The chosen factors and their three levels are shown in Table 1, and based on these data, the 27 experimental conditions were specified compared to 81 of a full factorial design. The notation of the synthesized ZnO-s is chosen by the value of the levels (−1, 0, 1) and according to the parameter order mentioned in Table 1 (e.g.: ZnO 1001 is ZnO synthesized at 190 °C, with a precursor concentration of 0.136 M in 60% ethanol-water mixture for 12 h).

The results were statistically analyzed using Minitab v.17 software. The relationship between the set of independent variables and the response (conversion, crystallinity) was evaluated based on the Box–Behnken design with a full quadratic model shown in the Equation (3).
Y = b_0_ + b_1_X_1_ + b_2_X_2_ + b_3_X_3_ + b_4_X_4_ + b_5_X_1_^2^ + b_6_X_2_^2^ + b_7_X_3_^2^ + b_8_X_4_^2^ + b_9_X_1_X_2_ + b_10_X_1_X_3_ + b_11_X_1_X_4_ + b_12_X_2_X_3_ + b_13_X_2_X_4_ + b_14_X_3_X_4_ + ε,(3)
where Y is the response (conversion, crystallinity), X_i_ (i = 1 to 4) are the independent variables and b_i_ (i = 0 to 14), are regression coefficients. 

ANOVA was applied to evaluate the quality of the model equation. The significance of model equation was statistically assessed by calculating the *p*-value (probability value—the probability of obtaining test results at least as extreme as the results actually observed during the test) with the significance level of 95% (*p* < 0.05). The model goodness of fit was evaluated by the coefficient of determination (R^2^) and the reproducibility of experimental data was determined just by errors. The validation of the model was tested by the generation of new input variables based on the model. In order to have a better overview of the model, response surfaces and contour plots were also generated.

## 3. Results and Discussion

In the case of all the samples the synthesis was performed with two autoclaves simultaneously with 150 and 300 mL internal volumes and 70% fill. Yet, the results of structural, optical, and photocatalytic activity investigations showed no observable differences, indicating that the process is not sensible of volumetric scaling in the respective range. Further on there is no need to discuss separately or in comparison the samples prepared in the two different autoclaves.

### 3.1. Characterization

Some of the representative XRD patterns of the synthesized ZnO samples can be seen in Figure 3. As may be noticed, the samples contained only the hexagonal wurtzite crystal phase (P63mc), which was evidenced from the identified diffraction peaks (JCPDS card no.0-3-0888) at scattering angles of 2θ: 31.40, 34.4, 36.3 corresponding to (1 0 0), (0 0 2), (1 0 1) crystallographic planes. Additional X-ray diffractograms can be found in the Appendix A. On the diffraction patterns, it could be observed that the ratio of the intensities of these peaks differs from sample to sample (a few examples could be found on Figure 3), yet in order to relate plausibly to the PDE, ANOVA was applied in function of the input parameters for the ratio of intensities of (0 0 2)/(1 0 0), which is discussed in later sections. Mean crystallite size was calculated, ranges 30–46 nm for the samples. The light absorption proprieties of the ZnO samples were determined by DRS and the band-gap values were calculated using Kubelka–Munk transformation and the first derivative of the spectra, yet no conclusive difference can be found amongst the samples, as all determined band-gap values range between 3.08–3.15 eV using Kubelka–Munk transformation and 3.15–3.24 eV using the first derivative, which is a too small range of variation, but confirms that our catalysts should be active in the UV-A region. The DRS spectra can be found in the Appendix A.

### 3.2. Photocatalytic Degradation Test

Methyl orange was chosen as model pollutant because it is the most widely used dye in testing the photocatalytic activity of various semiconductor catalyst [30], as well in the case of ZnO, because the photocatalytic degradation has proven sensible for material engineering of semiconductor photocatalyst, such as structure changes, surface proprieties, morphology, and light absorption proprieties [31,32]. All prepared samples have been proven to exhibit photocatalytic activity, as during photolysis experiments MO decolorization was <1 μM, the MO concentration does not change more than 2 μM, when the suspension is kept in dark and the change in the concentration is detectable only at the first sampling (after 10 min in dark); This assures that decolorization during irradiation is due to the influence of the catalysts.

As can be observed in Figure 4, the variation of the MO concentration shows strong linearity, as expected [33,34,35]. It is easy to observe that each experimental condition has an impact upon the photocatalytic activity; also, the ratio between the intensities of diffraction peaks associated with crystal facets (0 0 2), (1 0 0) changes accordingly, but in order to adequately present the relation between them, the response surface plots will be used for discussion.

### 3.3. Model Fitting and ANOVA Analysis

A fractional factorial design proves to be very effective in the present case because conducting experiments for just 27 sets of reaction coordinates is much more convenient to explore the effects of the parameters, as compared to the number of experiments required for a full factorial design (3^4^). This decrease in the number of experimental runs undoubtedly reduces the possibility of human error and saved 66% of the time and resources to conduct the synthesis and analysis. In general, the fractional experimental designs are sensible to the mathematical behavior of the output. To avoid interference or false-positive results from the experimental design, the PDE values were evaluated only in that time interval from the degradation, where the degradation curve could be considered linear (the fit of linear equation exceeded an R^2^ ≥ 97%), so degradation values at 40 min were selected for input PDE values. Equation (2) was fitted separately for the PDE, ratio of intensities (0 0 2)/(1 0 0), which was further evaluated by ANOVA analysis. The correlation resulted from the full quadratic equation fitting with their respective coefficient values are presented in Equation (4), respectively (5) and were further used to optimize the synthesis method to maximize the photocatalytic activity of ZnO.
PDE = −168.8 + 1.629X_1_ + 906.6X_2_ + 0.999X_3_ + 4.98X_4_ − 0.006818X_1_^2^ − 1232X_2_^2^ − 0.00558X_3_^2^ − 0.5951X_4_^2^ − 1.007X_1_X_2_ + 0.004417X_1_X_3_ + 0.01362X_1_X_4_ − 6.581X_2_X_3_ − 12.59X_2_X_4_ + 0.05979X_3_X_4_(4)
r_(002)/(100)_ = 0.854 − 0. 00404X_1_ − 1.28X_2_ − 0.00929X_3_ + 0.0107X_4_ + 0.000005X_1_^2^ + 8.94X_2_^2^ + 0.000117X_3_^2^ − 0.00214X_4_^2^ − 0.01471X_1_X_2_ + 0.000035X_1_X_3_ + 0.0003X_1_X_4_ + 0.00135X_2_X_3_ − 0.0083X_2_X_4_ − 0.000385X_3_X_4_,(5)
where, X_1_, X_2_, X_3_, and X_4_ are the process factors of the following parameters: temperature (°C), concentration of the precursor (M), ethanol content of the solvent (%) *v*/*v* of EtOH, and the duration of the solvothermal treatment (h), and PDE (%) and r_(002)/(100)_ (ratio of intensities of diffraction peaks corresponding to (0 0 2) and (1 0 0) crystallographic planes). As shown in Figure 5a, good agreement exists between the predicted results and those obtained from experiments for both responses. The ANOVA results of ZnO photocatalysts synthesis are shown in Table 2 and Table 3.

From the data presented in Figure 5, it becomes obvious that the parameters do show great variance upon the activity. One of the first measures for a good fit is the low standard deviation values (S) of 2.15(2) for PDE and 0.039(0) for the crystallographic ratio. However, low S values do not indicate properly that the model meets the model assumptions. R values is a basic statistical measure of how close the data are to the fitted values, which in present case, for both responses (PDE and r_(002)/(100)_) the R^2^ are 0.9913 and 0.9743, respectively, and are indicating the validity of the predicted PDE and r_(002)/(100)_; also, a relatively high value of the adjusted R^2^ coefficients (R^2^ adj is 0.9338 for PDE and 0.9443 for r_(002)/(100)_) was obtained, which means that the final prediction is in good agreement with the experimental results and accounts for 99% (PDE) and 97% (r_(002)/(100)_) of the variance [28,29]. ANOVA analysis reveals that all the studied factors are with a highly significant *p* value (*p* < 0.05) for PDE, but for the crystallographic parameter only half of the parameters exceeds the significance threshold. The Variance inflation factor (VIF, it is presented in the Appendix A) values are 1.00 or 1.25 in both cases, which confirms that the predictors are not correlated and there is no multicollinearity in none of the models, subsequently the predictors are stable. The lack of the fits’ significance is *p* = 0.054 (for PDE) and *p* = 0.059 (for intensity ratio) and are close to the threshold *p* value (0.050), which can be the consequence of the significance of the square terms, as it could suggest that some factors tend more toward non-linear behavior, as expected by a thermal process [25]. 

The significance of the terms in the full quadratic equation is much different in the case of ratio of intensities (0 0 2) and (1 0 0), as can be seen in Table 3, half of the terms shows significance. Linear and second order terms’ significance appears for the ethanol precursor concentration and temperature related interaction terms dominates. Taking in consideration that temperature and duration of solvothermal treatment only appear significant in interaction terms suggests that these parameters predominately produce impact during the main crystallization, implying the secondary crystallization is less occurrent [24].

The same model fitting was carried out for mean crystallite size of the samples. Unfortunately, the size showed no conclusive correlation with the synthesis parameters. This probably can be explained by the relatively narrow range, 30–46 nm, but the standard deviation is higher for 3 nm and R^2^ is less than 0.6, which makes it ineligible to uphold the proposed quadratic model.

### 3.4. Adequacy of the Regression Model

In order to optimize a higher PDE of ZnO photocatalysts by avoiding poor and undesired results, a fit of the experimental data was performed. Figure 6 shows all the diagnostic plots of ZnO optimization with both photocatalytic degradation efficiency and the ratio of the two mentioned crystallographic peaks to evaluate the adequacy of the regression model of prediction. From Figure 5, it can already be seen that both the PDE and the r_(002)/(100)_ values predicted by the fitted model are very close to the experimentally determined (the actual values are presented in Appendix A). Generally, the residuals must be evaluated as their distribution signify if the model is a real description of the mathematical behavior of the experimental data, which is revealed at Figure 6. In our case, one of the most significant observations at all diagnostic plots is that for both analyses the residuals are randomly distributed, as shown on the normal probability plots of residuals as all are situated close to the reference line, which represents a perfect normal distribution and residual vs. fitted value plots, so we could consider the errors present are independent and do not show skew or specific tendency out of randomness. These align with data presented in Appendix A as the histogram of the residuals shows a good symmetrical distribution, revealing no outlier run in the considered range.

From the standardized effects of the equation terms plot (Figure 6e,f) can be observed that the significance of 3 terms, which produce major effect upon the output follows the following order: for PDE X_3_ > X_1_^2^ > X_2_X_3_ and for r_(002)/(100)_ X_3_ > X_3_^2^ > X_2_. As it can be seen, the highest impact on both responses is the composition of the solvent applied during the solvothermal treatment, also worth mentioning that the duration falls beyond the other factors in significance, and the impact on both responses are much less, implying that crystallization process takes place relatively fast, the effect of these terms it is discussed further in alignment with the contour and response surface plots.

### 3.5. Effect of Synthesis Factors as Surface and Contour Plots

The effect of each factor on the synthesis of ZnO photocatalysts was investigated in a 3-D response surface and contour (2-D) graphs created using the full quadratic model. Figure 7 and Figure 8 show the effect of the different interactions between all the factors by varying two factors within the experimental ranges, while the other factors were held at the central point values (C with parameters: precursor concentration 0.136 M, 60% *v*/*v* ethanol-water solvent, 140 °C, 8 h solvothermal treatment). These effects were explained individually using statistical values, bringing more evidence on how the effects occurred, while varying the factors within specified ranges. Even though almost all of the terms of the fitted equation contribute significantly in case of PDE, we mentioned before, that terms involving ethanol concentration, temperature and precursor concentration constitute the predominant changes, the same is true for the r_(002)/(100)_, based on the paretto chart of standardized effects.

Based on standardized effects, the first order term of ethanol concentration (X_3_) generates the highest impact on the final activity of the synthesized photocatalyst, also on the r_(002)/(100)_. The generated surface and contour plots involving ethanol concentration (Figure 7) reveal that with the increase of ethanol concentration in the solvent the photocatalytic activity increase; as in all cases, samples synthesized using 90% ethanol solution are exhibited PDE values 2–4 times higher, compared to those synthesized at 30% ethanol, also this same observation can be clearly made for r_(002)/(100)_, as it proves to be this diffraction peak (0 0 2) becomes more dominant in a similar linear fashion as the PDE.

Hence, both these proprieties of the catalyst are mainly defined by this one parameter and the variance exhibits a very similar behavior, the synthesized ZnO photocatalytic activity is mainly controlled by the solvent composition, which controls distribution of crystallographic facets and in parallel simultaneously with the activity of the prepared catalyst. It has already been proven that organic solvents lead to higher photocatalytic activities in case of solvothermal [36] and sol-gel synthesis [37], because the formation of oligomers from zinc-acetylacetonate hydrate is extensively more controllable by the composition of the solvent [38], due to its higher solubility in organic solvents. In addition, zinc-acetylacetonate hydrate is highly reactive towards water, leading to undesired Zn(OH)_2_, which would interrupts the slow process of formation of oligomers, their dehydration to form ZnO [39], inherently preventing a well-controlled crystallization in this temperature intervals [11]. 

As we further examine the surface plots which involve the precursor concentration (Figure 7 and Figure 8), we can conclude that lower concentration of the precursor always leads to higher activity. This is especially obvious in Figure 7, which strongly strengthens that as the ethanol content is also a defining parameter through synergetic effects for both responses (at 30% ethanol the PDE and r_(002)/(100)_ varies only between 25 and 45%, and 0.24 and 0.36, respectively, as at 90% ethanol these are more than doubled) [40]. Due to the simple fact the more precursor is present, it becomes more probable that the initial crystallization reactions will take place at a faster rate, leading to less controllable process.

On the other hand, parameters representing the solvothermal treatment (temperature, X_1_ and duration, X_4_) breaks the similarity between the behavior of the two responses, because as the (0 0 2) peaks intensity would increase generally with the temperature, as for the PDE the plots (Figure 8) suggest optimum temperature inside the intervals examined, but for the ratio of intensities the maximum is at the boundaries of the system. It is worth mentioning that the crystallization process of ZnO is strongly activated and governed by the temperature, because it involves endotherm reactions [17]. Thus, the final steps of dihydroxylation and deprotonation should exhibit a more exponential behavior, which is clearly observable on Figure 8 and Appendix A. Even though duration of the solvothermal treatment is not considered with high significance compared to the previous 3 parameters and does not inflict greater changes to the photocatalytic activity, showing neither strong individual nor synergetic effects. Yet, the surface plots of the two models suggests an optimum inside examined parameter intervals, presuming that the shorter synthesis would suffice for a desired optimum activity, which has economically benefitted.

The (0001) crystal facet (corresponds to (0 0 2) diffraction peak) has been proven that could directly determine the photocatalytic activity of ZnO [36,41] and to our best knowledge the literature focused more on much higher ratios (r_(002)/(100)_ > 1), where this ratio of intensities in the case of methylene blue [42] followed the change in activity of ZnO catalyst, but had a negative impact on the photoinduced reaction, also the same trend was observable for the degradation of rhodamine B [43,44,45] and photoreduction of CO_2_ [46], which is opposite to the present case with methyl orange. In addition, ZnO with intensity ratios of (1 0 0)/(0 0 2) in the present region, synthesized with similar methods exhibit better photocatalytic activities, even though that research does not focus on this aspect. Considering that in the previously mentioned literature does not discuss the present interval of ratio of these crystallographic facets (0.2–0.9) proves that this interval could induces an opposite trend upon the photocatalytic activity.

### 3.6. Optimization and Model Validation

The main objective of the present research was to optimize the synthesis variables for the preparation of ZnO photocatalyst with highest photocatalytic activity in the examined intervals of parameter values. r_(002)/(100)_, in terms of the two main parameters, showed similar trends as PDE.

Therefore, the optimization was carried out for this response too, in order to reflect this aspect’s relation upon the activity. Minitab software was selected to determine the maximum value for both r_(002)/(100)_ and PDE separately, and the obtained parameter values were implemented in new experiments to test the validity of the model. Although, we need to consider the previously examined surface plots, which revealed that two of these parameter values are at limits of the investigated interval, which would mean that the model would be tested partially with data already applied as a starting data set, so another set of experimental data was randomized to ensure full extent testing of the variables [44]. The result of the optimization and validation experiment is presented in Table 4. As expected, the “optimum” values of the synthesis parameters intersect at two most defining parameters, the ethanol concentration and precursor concentration with exact match: 90% ethanol, 0.068 M ZnAA_2_ concentration.

Yet, these values are at the boundaries of the model, implying an even higher maximum outside of the proposed intervals. Hence, these two parameters coincide; we could generate the surface plot for temperature and time at the hold values of 90% ethanol and 0.068 M precursor concentration (Figure 9) and reveal that, amongst the parameters of solvothermal treatment, time does not impose great impact, rather the temperatures and its synergetic effect with time. There is a difference between the optimum temperature for PDE and r_(002)/(100)_ models, but in this interval (156–190 °C), none of these responses change significantly. This experimental result supports, by strong evidence, that the BBD model selection and applied methodology was sufficiently good, as predicted values are close to the experimental ones for both PDE and r_(002)/(100)_; yet, this model’s main purpose is to discover the main parameters and their possible behavior. The samples photocatalytic activity was compared to a commercial photocatalyst Degussa p25 titanium dioxide (P25), and the activity of the optimized ZnO sample is almost identical to it, as can bee seen in Figure 10.

The reusability of the catalyst is an important criterion for a catalyst for industrial application [47]. The best performing catalyst is the PDE optimized ZnO (ZnO-PDE opt) and the stability of the sample was investigated through the reuse of the catalyst in the degradation experiment. In Figure 11 can be seen that the methyl orange conversion decreases around 1% each cycle. A part of this decrease can be attributed to material losses from the operations between cycles, so the catalyst can be considered stable.

### 3.7. The Relation of (0 0 2) Peak to the Photocatalytic Activity of ZnO

At first, the present study shows a little controversy to some of the existing literature discussing the relation of photocatalytic activity of ZnO to the observed proportion of the X-ray diffraction peak corresponding to (0 0 2) crystal plane. A proposed growth of ZnO crystal along (002) crystallographic plane is represented in Figure 12a. This would indicate that particles with similar size and high (002)/(100) XRD peak ratio should also prefer a shorter hexagonal bar like morphology with pyramidal tip. Even though, in the beginning of present works the aim was to exclude electron microscopy investigation, in Figure 12b,c there is two samples with different r_(002)/(100)_. The micrographs confirm the bar-like conic morphology, but XRD results offered a better explanation for the trend in activity, as based on the micrographs there is no observable difference between the two samples, but the difference in PDE is noticeable (10%).

Beforehand, we mentioned that there are some publications which support the exact opposite of the present works experimental data, that the lower the ratio of X-ray diffraction peak (0 0 2) to (1 0 0), the higher the catalytic activity should increase. Yet, these works aim at much greater (>1.2, up to 5) or lower ratio (<0.2) [45]. Most studies do not focus the present studies interval (0.2–0.9) and a large number of articles discuss the synthesis of ZnO, but this interval of crystallographic ratio is omitted. Even though, this data would show a trend similar to the observed one in present study. A few of these types of examples are presented in Table 5.

In literature, it has been demonstrated that some crystal facets show preference for photocatalytic reactions and different light harvesting abilities, but these assumptions should not be polarized to the extreme that one crystal facet should be supreme to an another in a photocatalytic point of view. Usually, a change in intensity ratios on X-ray diffractograms is a sign that the crystal growth is influenced and the crystal facet corresponding to the specific reflection is more likely to be induced by a dominant facet on the boundaries of the crystallites. (0 0 2) diffraction corresponds to crystal growth alongside (0001) and (0001^−^) direction of ZnO. This crystal facet is a polar facet and it has been reported that high concentration of water could inhibit the growth alongside this direction, the terminal Zn^2+^ and O^2−^ along (002) crystal plane can strongly interact with the polar water molecules (Figure 12a), respective to its dissociated forms (OH−, H+) [20,34,40]. This explains why in present case the higher ethanol concentration is the primary controlling parameter of this facet. As the ethanol content is increased and implicitly the water content is decreased, less adsorption of water molecules can occur on the specific crystallographic planes. It was demonstrated several times in the literature that these facets’ polar nature contributes predominantly to the photocatalytic efficiency of ZnO, but is not the only instigator of catalytic processes. As it seems there is an optimum in synergetic effects of this crystal facets as supported by our present study.

## 4. Conclusions

Based on 3 level Box–Behnken design, the 4 major determining parameters of solvothermal synthesis of ZnO has been investigated: temperature (90, 140 190 (°C)), concentration of the precursor in the reaction mixture (0.068, 0.136, 0.204 (M)), ethanol content of the solvent (30, 60, 90 (%) *v*/*v*) and the duration of the solvothermal synthesis (4, 8, 12 (h)). The obtained samples the photocatalytic degradation efficiency of MO was measured under UV irradiation and were characterized using XRD measurements. A full quadratic model was fitted between investigated parameter and 2 responses, the photocatalytic degradation efficiency and the ratio of intensities of two diffraction, corresponding to two polar crystal facets (0 0 2) and (1 0 0). The model adequacy was validated by ANOVA analysis obtaining a good correlation, as proven by R^2^ of 0.9913 for PDE. and 0.9743 for r_(002)/(100)_, which indicates the validity to the predicted PDE.

Obtained models were successfully used to optimize the synthesis parameters and the obtained optimum parameters were validated by new experimental determinations. The computed optimal parameters led to a 88.0 (%) conversion of MO for the ZnO, value obtained for the optimal parameters of the solvothermal synthesis: temperature of 154 °C, concentration of the precursor ZnAA_2_ in the reaction mixture of 0.068 (M), ethanol content of the solvent of 90 (%) *v*/*v* ethanol and time of the synthesis of 9.74 (h). Nevertheless, with the obtained models and the validation experiments we were able to demonstrate that, with this synthesis and model, the activity of the obtained ZnO can be tailored along with a crystallographic aspect, the ratio of intensities of facets (0 0 2) and (1 0 0) in the range of 0.2–0.88. Therefore, this study showed that reduced experimental designs can be used for both the photocatalytic and crystallographic design of ZnO.

## Figures and Tables

**Figure 1 nanomaterials-11-01334-f001:**
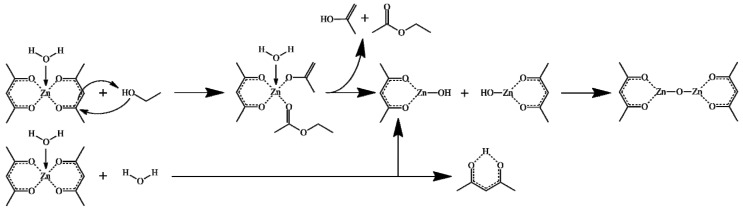
Proposed mechanisms of Zn-O-Zn bond formation from zinc-acetylacetonate monohydrate. 2.3. Characterization.

**Figure 2 nanomaterials-11-01334-f002:**
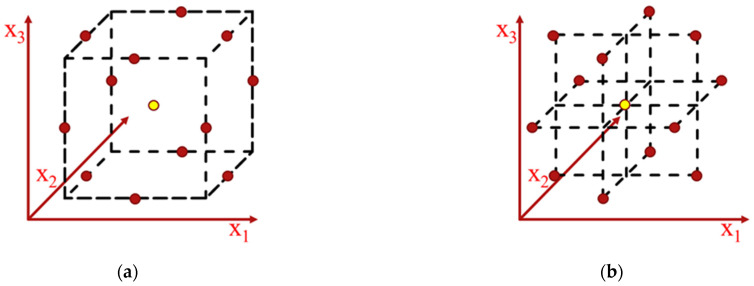
The forms of Box–Behnken design: (**a**) cube; (**b**) three interlocking 3 factorial design.

**Figure 3 nanomaterials-11-01334-f003:**
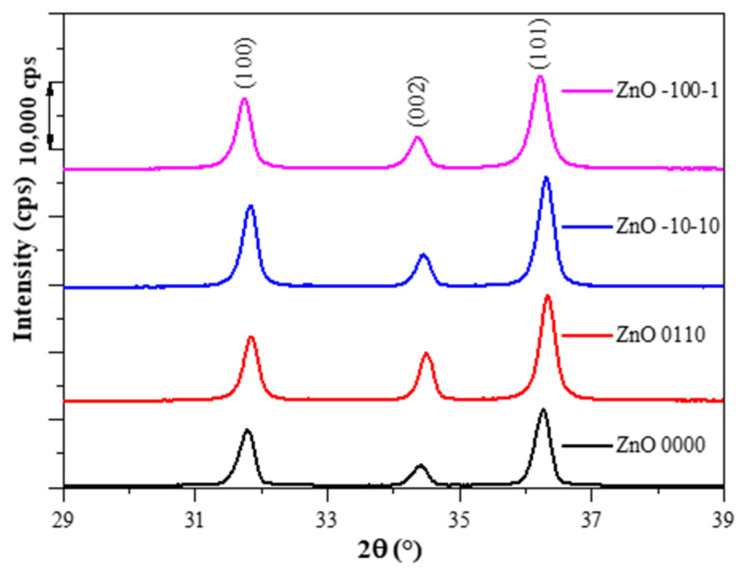
XRD patterns of some representative ZnO samples showing the wurtzite crystal phase.

**Figure 4 nanomaterials-11-01334-f004:**
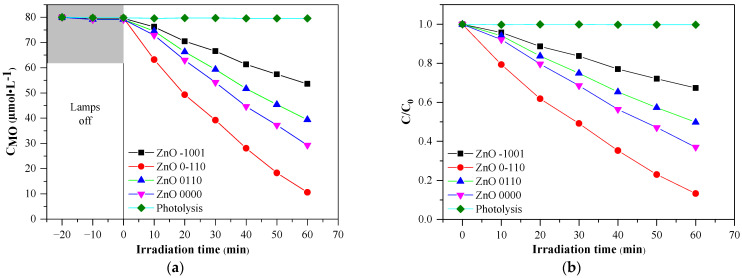
The MO degradation curves of some representative ZnO samples: (**a**) MO concentration vs. time; (**b**) C/C_0_ vs. time.

**Figure 5 nanomaterials-11-01334-f005:**
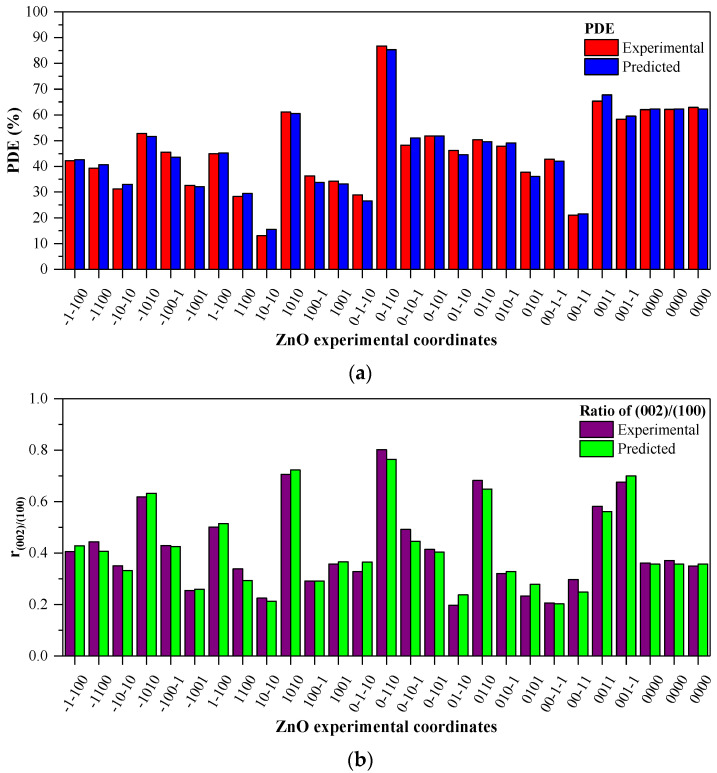
Experimental runs of Box–Behnken design with the comparison between predicted and experimental photocatalytic degradation efficiency: (**a**) in case PDE; (**b**) in case ratio of intensities (0 0 2)/(1 0 0).

**Figure 6 nanomaterials-11-01334-f006:**
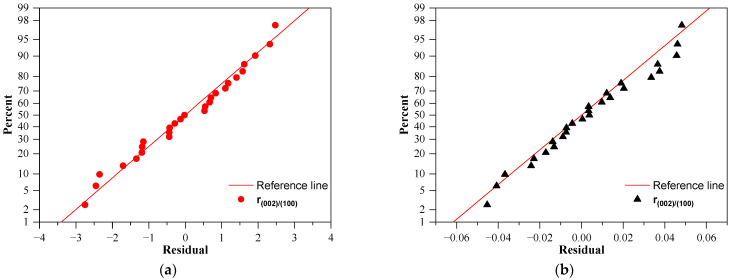
Diagnostic plots of ZnO synthesis results: normal probability plot of residual: (**a**) for PDE; (**b**) for r_(002)/(100)_ and residual vs. fitted values: (**c**) for PDE; (**d**) for r_(002)/(100)_; standardized effects of the equation terms (**e**) for PDE; (**f**) for r_(002)/(100)_.

**Figure 7 nanomaterials-11-01334-f007:**
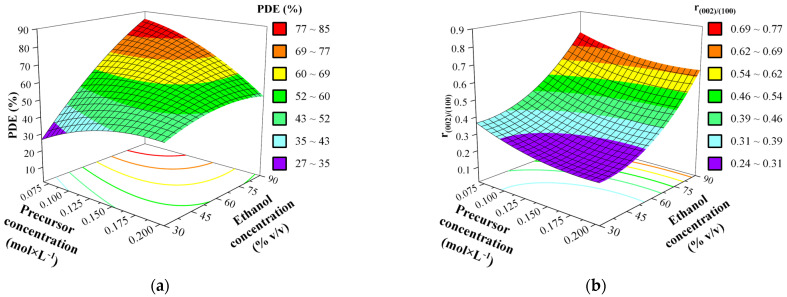
Effect of interaction between precursor concentration (X_2_) and ethanol content of the solvent (X_3_) (**a**) on the PDE and (**b**) r_(002)/(100)_ of ZnO as 3D response surface. Fixed parameter values: 140 °C, 8 h solvothermal treatment.

**Figure 8 nanomaterials-11-01334-f008:**
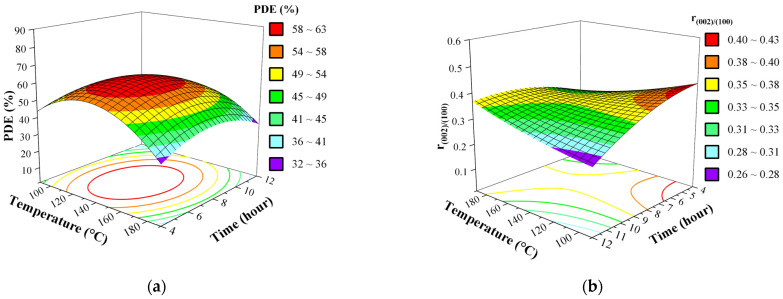
Effect of interaction between temperature (X_1_) and duration (X_4_) of the solvothermal treatment (**a**) on the PDE and (**b**) r_(002)/(100)_ of ZnO as 3D response surface. Fixed parameter values: precursor concentration 0.136 M, 60% *v*/*v* ethanol-water solvent.

**Figure 9 nanomaterials-11-01334-f009:**
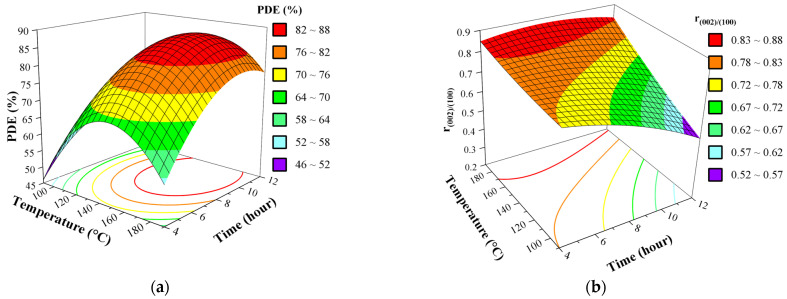
Surface plots of time and temperature of the solvothermal treatment in the optimization region of parameter intervals (at hold values 90% EtOH and 0.068 M ZnAA_2_) for (**a**) PDE and (**b**) r_(002)/(100)_. Fixed parameter values: precursor concentration 0.068 M, 90% *v*/*v* ethanol-water solvent.

**Figure 10 nanomaterials-11-01334-f010:**
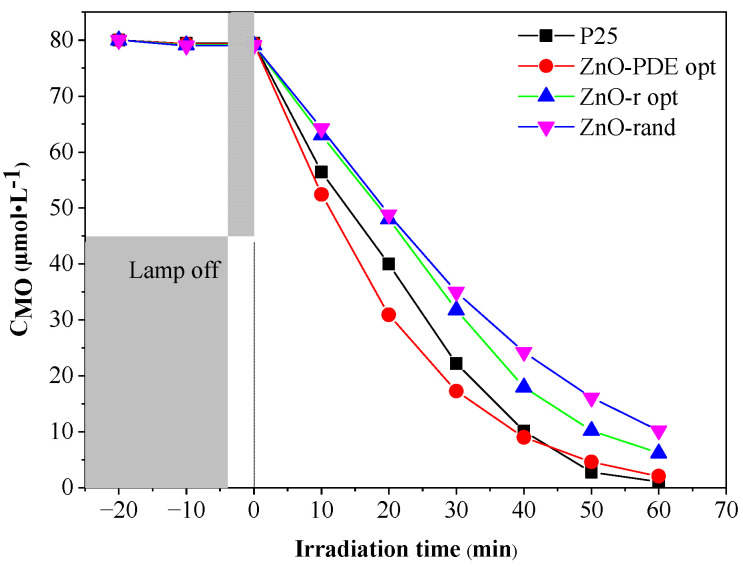
The MO degradation curves of the validation experiments of ZnO.

**Figure 11 nanomaterials-11-01334-f011:**
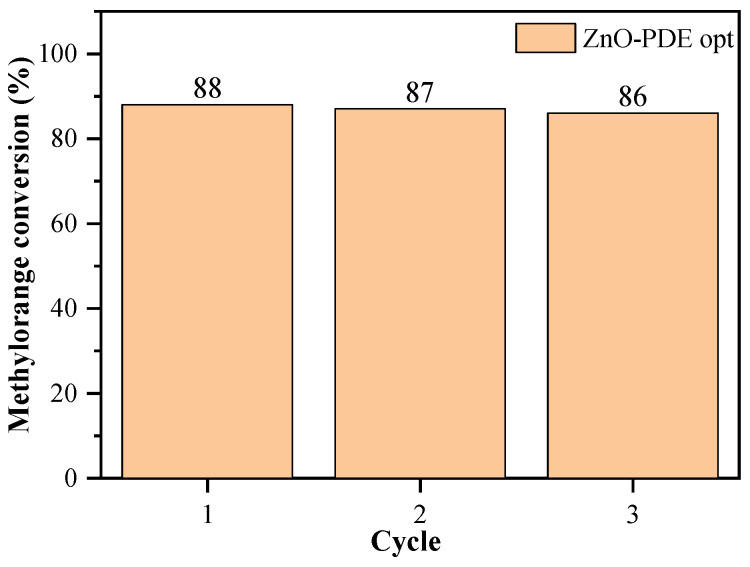
The reusability test for the best performing sample, ZnO-PDE opt.

**Figure 12 nanomaterials-11-01334-f012:**
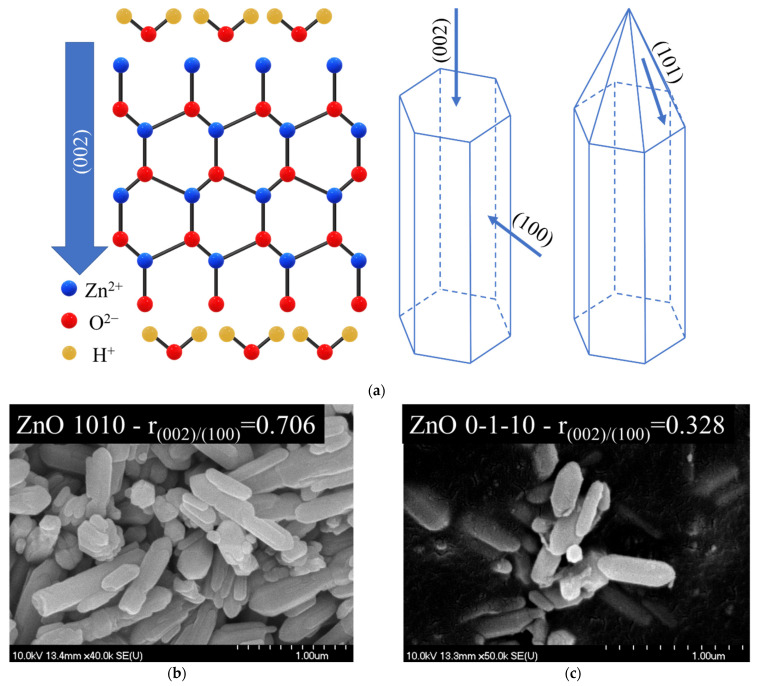
(**a**) Schematic representation of growth habit of ZnO along different crystallographic planes; Scanning electron micrographs of samples (**b**) ZnO 1010; (**c**) ZnO 0-1-10.

**Table 1 nanomaterials-11-01334-t001:** Variables, levels of variables and constrains used for Box–Behnken design.

Factors	Symbol	Levels		
		−1	0	1
Temperature [°C]	X_1_	90	140	190
Concentration of the precursor [M]	X_2_	0.068	0.136	0.204
Ethanol content of the solvent [% *v*/*v*]	X_3_	30	60	90
Duration of the solvothermal treatment [h]	X_4_	4	8	12

**Table 2 nanomaterials-11-01334-t002:** ANOVA results for quadratic model of ZnO using Box–Behnken design for photocatalytic measurements.

Analysis of Variance					
Source	DF	Adj SS	Adj MS	F-Value	*p*-Value
Model	14	6326.51	451.89	97.32	<10^−3^ *
Linear	4	3443.08	860.77	185.37	<10^−3^ *
X_1_	1	55.04	55.04	11.85	<10^−3^ *
X_2_	1	234.97	234.97	50.6	<10^−3^ *
X_3_	1	3043.27	3043.27	655.37	<10^−3^ *
X_4_	1	109.81	109.81	23.65	<10^−3^ *
Square	4	1657.47	414.37	89.23	<10^−3^ *
X_1_^2^	1	1549.66	1549.66	333.72	<10^−3^ *
X_2_^2^	1	173.03	173.03	37.26	<10^−3^ *
X_3_^2^	1	134.45	134.45	28.95	<10^−3^ *
X_4_^2^	1	483.45	483.45	104.11	<10^−3^ *
2-Way Interaction	6	1225.96	204.33	44	<10^−3^ *
X_1_ × X_2_	1	46.92	46.92	10.1	<10^−3^ *
X_1_ × X_3_	1	175.56	175.56	37.81	<10^−3^ *
X_1_ × X_4_	1	29.7	29.7	6.4	<10^−3^ *
X_2_ × X_3_	1	720.92	720.92	155.25	<10^−3^ *
X_2_ × X_4_	1	46.92	46.92	10.1	<10^−3^ *
X_3_ × X_4_	1	205.92	205.92	44.35	<10^−3^ *
Error	12	55.72	4.64		<10^−3^ *
Lack-of-Fit	10	55.08	5.51	17.03	0.057 **
Pure Error	2	0.65	0.32		
Total	26	6382.23			

* significant (*p* < 0.05); ** not significant; S = 1.83, R^2^ = 0.9913, R^2^(adj) = 0.9811, R^2^(pred) = 0.9501.

**Table 3 nanomaterials-11-01334-t003:** ANOVA results for quadratic model of ZnO using Box–Behnken design for the ratio of intensities of X-ray diffraction peaks.

Analysis of Variance					
Source	DF	Adj SS	Adj MS	F-Value	*p*-Value
Model	14	0.693	0.050	32.5	<10^−3^ *
Linear	4	0.557	0.139	91.31	<10^−3^ *
X_1_	1	0.001	0.001	0.38	0.551 **
X_2_	1	0.044	0.044	28.98	<10^−3^ *
X_3_	1	0.506	0.506	331.72	<10^−3^ *
X_4_	1	0.006	0.006	4.17	0.064 **
Square	4	0.092	0.023	15.16	<10^−3^ *
X_1_^2^	1	0.001	0.001	0.52	0.484 **
X_2_^2^	1	0.009	0.009	5.98	<10^−3^ *
X_3_^2^	1	0.059	0.059	38.92	<10^−3^ *
X_4_^2^	1	0.006	0.006	4.09	0.066 **
2-Way Interaction	6	0.044	0.007	4.85	<10^−3^ *
X_1_ × X_2_	1	0.010	0.010	6.56	<10^−3^ *
X_1_ × X_3_	1	0.011	0.011	7.44	<10^−3^ *
X_1_ × X_4_	1	0.014	0.014	9.45	<10^−3^ *
X_2_ × X_3_	1	0.000	0.000	0.02	0.89 **
X_2_ × X_4_	1	0.000	0.000	0.01	0.91 **
X_3_ × X_4_	1	0.009	0.009	5.61	<10^−3^ *
Error	12	0.018	0.002		<10^−3^ *
Lack-of-Fit	10	0.018	0.002	16.37	0.059 **
Pure Error	2	0.000	0.000		
Total	26	0.712			

* significant (*p* < 0.05); ** not significant; S = 0.039, R^2^ = 0.9743, R^2^(adj) = 0.9443, R^2^(pred) = 0.8531.

**Table 4 nanomaterials-11-01334-t004:** The optimization of the ZnO synthesis for optimum photocatalytic degradation efficiency.

Parameters	Temp.	Precursor Conc.	EtOH Conc.	Duration	Predicted		Experimental	
	[°C]	[M]	[% *v*/*v*]	[h]	PDE	r_(002)/(100)_	PDE	r_(002)/(100)_
					[%]	[a.u.]	[%]	[a.u.]
Optimized r_(002)/(100)_ (ZnO-r opt)	190	0.068	90	7.56	74.90	0.873	77.30	0.834
Optimized PDE (ZnO-PDE opt)	155	0.068	90	9.75	88.03	0.766	89.10	0.799
Randomized (ZnO-rand)	107	0.102	69	9.00	67.18	0.516	69.30	0.534

**Table 5 nanomaterials-11-01334-t005:** Comparison of literature data of different ZnO photocatalysts based upon observed ratio of (0 0 2)/(1 0 0) and activity.

Synthesis Method	Photocatalytic Activity Experiment	Range of (0 0 2)/(1 0 0) Variation	Observed Trend of Photocatalytic Activity with the Ratio	Reference
Solvothermal	Photoreduction CO_2_	0.91–1.27	Increased	[46]
Chemical bath deposition process	Photodegradation of Rhodamine B	1.28–1.9	Decreased	[41]
High temperature chemical precipitation		0.63–0.92	Increased	[43]
Chemical precipitation	Photodegradation of phenol	0.54–0.93 (I), 0.93–1.13 (II)	I-Increased, II-Decreased	[48]
Electrospinning deposition	Photodegradation of Rhodamine B	0.81–1.06	Increased	[49]
Combustion	Methylene Blue, Crystal Violet	0.4–0.55	Increased	[50]
Solvothermal	Methylene Blue	1–5	Decreased	[44]

## Data Availability

Data is contained within the article or Appendix A.

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
