# Peer review of "Optimization Method of the Solvothermal Parameters Using Box–Behnken Experimental Design—The Case Study of ZnO Structural and Catalytic Tailoring"

_nanomaterials, 2021, doi:10.3390/nano11051334_

Round 1

Reviewer 1 Report

I recommend to improve the characterization of the materials by providing SEM and TEM images.

Reviewer 2 Report

In this manuscript, Lucian Baia et al. investigated the 4 major determining parameters of solvothermal synthesis of ZnO and their effect on photocatalytic degradation activities. Although there are still some deficiencies in experimental data and mechanism explanation, I recommend it to be accepted by Nanomaterials after minor revisions. The authors should carefully consider the following comments:

  1. Please explain why the midpoint and range of the parameters studied in this paper were selected.
  2. As shown in Figure 2, the XRD patterns of ZnO prepared under different parameters obviously shifted, which should be explained in the manuscript.
  3. For the mainly discussed dependent variables, the correlation between the PDE and ratios of (002)/(100) was not very good. For example, there were several experiments in which the PDEs were high while the ratios were low (coordinates of 0000, 00-1-1, 01-10). Please indicate if there are other factors.
  4. Figure 6 and 7 showed the effect of two parameters on the PDE and r(002)/(100) of ZnO. However, the values of the other two parameters were not clearly mentioned. It is suggested to add the values of fixed parameters to the notes in Figure 6 and 7.
  5. In Table 4, the experimental r(002)/(100) was -0.799 while the predicted was 0.766. Please confirm that.
  6. Some very related works could be useful for the discussion like Small Struct. 2021, 2, 2000061; Rare Met., 2019;38(4):277-286;

Reviewer 3 Report

-The abstract should state briefly the purpose of the research, the principal results and major conclusions. The authors should rewrite the abstract for these standards.

-detect the chemical composition stability of the samples after used in catalytic process

-Introduction should be enhanced by recently references as; doi.org/10.3390/molecules26082269

-add error bar in application curves and for the tables

-check the catalytic activity of prepared sample after reused for several times.

Reviewer 4 Report

Dear Authors, in your interesting manuscript, the following points should be added/changed to further improve it:

  1. Introduction: I suggest completing the description of the state of art about the solvothermal synthesis of nano ZnO. What about methods for controlling particle size of ZnO based on solvothermal synthesis ? In the review of the state of art, it is worth noting that the effect of changing the water content in solvothermal synthesis of ZnO has already been investigated for zinc acetate and ethylene glycol model (DOI:10.1155/2016/2789871). Mechanism of zinc oxide nanoparticle size control, which enables the size control of ZnO NPs obtained in microwave solvothermal synthesis within the size range between circa 20 and 120 nm through the control of water content in the solution of zinc acetate in ethylene glycol has been explained [DOI:10.1088/1361-6528/aaa0ef]. The effect of the ZnO nanoparticles size on the photocatalytic degradation of phenol in a water solution under the influence of UV and Vis has been investigated, where for the first time, research on photocatalytic degradation has used ZnO NPs produced by only one method (microwave solvothermal synthesis without heat treatment or other processes of reduction/oxidisation of the surface of NPs samples) [DOI:10.1016/j.apsusc.2020.148416]. I suggest also read the review article describing, among others, the state of art regarding the microwave solvothermal synthesis of zinc oxide nanomaterials, which was published in Nanomaterials in 2020. I am convinced that, as for the aforementioned articles, for solvothermal synthesis using zinc-acetylacetonate monohydrate and ethanol, the decisive reaction for the formation of ZnO will be the esterification reaction. Now the question is: have the authors noticed the same effect described in the literature?
  2. Introduction: Please explain what "different water" means in the sentence [72-75].
  3. Introduction: Comment to the sentence “Also, several precursors have been studied to prepare ZnO using solvothermal method such as inorganic salts, as well as organic complexes.“ Please provide examples of such works (references).
  4. Introduction: No explanation was given for the abbreviation “ANOVA” [123]
  5. Materials and Methods - Reagents and materials: Please provide information about specific conductivity of ultrafiltered water. Was it deionized water?
  6. Materials and Methods - Solvothermal synthesis of ZnO: Please explain to me how to understand the ethanol content expressed as a percentage by volume “X2 (30, 60 and 90% v/v). [139]” ? I would like to point out that there is no principle of preserving the volume. Please explain how calculated of the corresponding amount of MQ water [141].
  7. Materials and Methods - Solvothermal synthesis of ZnO: I suggest adding a chemical reaction equation representing the obtaining of ZnO. Whether one of the reaction products is esters ?
  8. Materials and Methods - Solvothermal synthesis of ZnO: I suggest adding information about the number of received samples and refer to the contents of Table S1.
  9. Materials and Methods – Characterization: No explanation was given for the abbreviation “PDE” [154].
  10. Results and discussion – Characterization: Why did the authors not calculate crystallite sizes from XRD data ?. Maybe the size effect will be visible here.
  11. Results and discussion: I propose to add a figure showing the growth habit of ZnO crystal. This will help readers make the discussion easier to understand .
  12. Results and discussion - Effect of synthesis factors as surface and contour plots: I don't understand part of the sentence "... as in mixture of water ethanol get more concentrated in ethanol, the less dissociation of water molecules occur in the reaction mixture. [502-503]".
  13. Results and discussion: What about the influence of the water content on the ZnO crystallization process ?.
  14. Results and discussion: My comment on the sentence “This crystal facet is a polar facet and it is been reported that high concentration of OH- present in the reaction mixture could inhibit the growth alongside this direction [31], which explains why in present case the higher ethanol concentration is a primary control of the  this facet, as in mixture of water ethanol get more concentrated in ethanol, the less dissociation of water molecules occur in the reaction mixture. [498-503]”. The publication [31] discusses the hydrothermal synthesis of zinc oxide, where the source of OH- is NaOH. We are talking about completely different OH- concentrations in solvothermal synthesis, where OH- concentration results from dissociation constants. I propose to conduct this discussion based on the chemical reaction equation for selected reactants (zinc-acetylacetonate monohydrate, absolute ethanol, water). It is worth establishing, even purely hypothetically, what is the role of water. Is water a reagent or a solvent, or maybe both?

Reviewer 5 Report

The optimization method has been convinsingly applied for photocatalytic optimization of ZnO.
The English is good in general, however there are some strange sentences. The authors should go through the manuscript and improve parts of it. Some examples of strange sentences: 
Page 12, row 494 "There is already been proven..."
page 16,row491: "..there is long been demonstrated that..."

Author Response

The optimization method has been convinsingly applied for photocatalytic optimization of ZnO.

The English is good in general, however there are some strange sentences. The authors should go through the manuscript and improve parts of it. Some examples of strange sentences:

Page 12, row 494 "There is already been proven..."

page 16,row491: "..there is long been demonstrated that..."

Answer

Thank you for the remark. The English of the manuscript has been improved, such as grammatical corrections, improper word order, etc.. These are non-scientific changes and do not change the discussion of the results, so are not detailed in the answer.

Round 2

Reviewer 3 Report

Accept in present form

Reviewer 4 Report

The answers from authors and the revised manuscript is acceptable at present form.